**Beyond tipping points: risks, equity and the ethics of intervention**
Laura M. Pereira 1, 2, , Steven R. Smith 3, 4 , Lauren Gifford 5 , Peter Newell 6 , Ben Smith 7 ,
Sebastian Villasante 8 , Therezah Achieng 3 , Azucena Castro 2, 9 , Sara M. Constantino 10 ,
Tom Powell, 3, Ashish Ghadiali 3 , Coleen Vogel 1 , Caroline Zimm 11
1 Global Change Institute, University of the Witwatersrand, Johannesburg, South Africa
2 Stockholm Resilience Centre, Stockholm University, Stockholm, Sweden
3 GSI, University of Exeter, UK
4 CUSP, University of Surrey, UK
5 Department of Ecosystem Science and Sustainability, Colorado State University, USA
6 University of Sussex, UK
7 Department of English, University of Exeter, UK
8 EqualSea Lab-CRETUS, University of Santiago de Compostela, Spain
9 Stanford University, Stanford, USA
10 Northeastern University, USA
11 International Institute of Applied Systems Analysis, Vienna, Austria
Correspondence to: Laura M Pereira (laura.pereira@wits.ac.za)
**Abstract**
Earth system tipping points pose existential threats to current and future generations, both
human and non-human, with those least responsible for causing them facing the greatest risks.
'Positive' social tipping points (that we shorten to positive tipping points, or PTPs) are often
deliberate interventions into human systems with the aim of rapidly mitigating the risks of Earth
system tipping. However, the desire to intervene should neither increase risks nor perpetuate
unjust or inequitable outcomes through the creation of sacrifice zones. In this paper, we argue
that considerations of what needs to change, who is being asked to change and where and by
whom the change or its impacts will be felt are fundamental and normative questions that
require reflexivity and systemic understanding of decision-making across scales. All actors have
a role to play in ensuring that justice, equity and ethics are carefully considered before any
intervention. Enabling positive tipping points for radical transformations would thus benefit from
more diverse perspectives, with a particular emphasis on the inclusion of marginalised voices in
offering solutions. We conclude that taking a cautious approach to positive tipping interventions,
including careful consideration of distributional and unintended consequences, and stepping
back to explore all options, not just those appearing to offer a quick fix, could lead to more
equitable and sustainable outcomes.

500 character blurb
Earth system tipping points pose existential threats requiring urgent action. However, this
imperative should neither increase risks nor perpetuate injustices. We argue that considerations
of what needs to change, who is asked to change and where the impacts will be felt and by
whom, are fundamental questions that need to be addressed in decision-making. Everyone has
a role to play in ensuring that justice and equity are incorporated into actions towards a more
sustainable future.
**1. Introduction**
The world is facing a series of era-defining, existential threats including climate change,
biodiversity loss, increased inequality and poverty. In response to these critical challenges,
there have been calls for transformative change (IPBES, 2019). Some of these transformations
are proposed as advancing 'positive' social tipping points, which we shorten to positive tipping
points (PTPs). PTPs are defined as changes to a system that  become self-perpetuating beyond
a threshold, and which lead to substantial, often abrupt impacts that are predominantly
beneficial to humans and the natural systems we rely on (McKay et al., 2022; Milkoreit et al.,
2018). As we argue, 'positive' is a value judgement, and not all the changes associated with
PTPs are universally welcome; difficult decisions and trade-offs need to be made as we weigh
up the distribution of anticipated harms and benefits. Nevertheless, we argue that there is a
collective duty to bring about "intentional transformation towards global sustainability" (Lenton et
al., 2022: 2), and this is clearly a normative enterprise. The moral 'force' in our usage of the
'positive' descriptor is based on the science of Earth system boundaries and the ethics of Earth
system justice (Gupta et al., 2023a; Rockström et al., 2023).
However, undertaking or operationalizing such transformations that attempt to orient complex
systems onto more safe and just trajectories is messy and complicated (Olsson and Moore,
2024). As history shows, there are dark sides of transformations, with unintended
consequences, distributional impacts and the potential for vested interests to co-opt or reap the
benefits of such processes (Blythe et al., 2018). Caution and care is thus necessary when
considering the use of PTPs, including clarity about what transformations are intended, whom
they benefit, and whom they may harm (Pereira et al., 2024).
Any moment of societal change will inevitably generate winners and losers (O'Brien and
Leichenko, 2003), and this should also be taken into account in the identification and
operationalisation of PTPs, where the aim is often to create both rapid and radical change.
Indeed, in this context, the language of positive tipping needs to be exercised with caution since
the very definition of a PTP is likely to be experienced by many actors as a polarising event and
can have differential welfare impacts on different segments of the population (Ehret et al.,
2022). For example, while some welcome a tipping point away from a fossil fuel-based economy
towards one dominated by renewables, (IEA, 2022b; IRENA, 2022; Systemiq, 2023), others in
fossil fuel and related industries may fear the loss of their livelihoods and communities.
Pollution, habitat destruction and poor working conditions in the expansion of cobalt and lithium
mining for battery production, for example, driven by the rapid increase in the production of
electric vehicles, may create problems for some communities and opportunities for others
(Hernandez and Newell, 2022).
An approach to tipping point governance that centres principles of equity and justice (Okereke
and Dooley, 2010) will recognise that tipping points, whether conceived primarily as positive or
negative, will leave segments of the population behind without the engagement of
complementary redistribution mechanisms that can help mitigate against the worst impacts of
change (Rammelt et al., 2023). This paper is not proposing how to govern tipping points, but
rather focuses on the equity and justice challenges that are often overlooked in discussions of
both Earth system and social tipping points. When identifying or triggering a tipping point
through an intervention, it is necessary to ask: What kind of trade-offs are necessary and what
sacrifice zones are being created? Who ends up occupying these sacrifice zones? Who is left
behind? And how can a comprehensive understanding of justice be included in a rigorous way
when examining PTPs? An example of sacrifice zones are extractive zones created by the
advancement of coordinated forms of capitalism that see those territories and the communities
inhabiting them as commodifiable (Gómez-Barris, 2017).

## 96  1.1. Climate Justice in light of Tipping Points
Recent UNFCCC climate summits have seen increasing calls from climate justice campaigners
and representatives of the Global South, including the small island developing states, for a
global recognition of the uneven historical and ongoing responsibility for climate change,
articulated in the concept of "common but differentiated responsibilities" and calls for 'loss and
damage' and elsewhere for reparations (Constantino et al., 2023; Huq et al., 2013). These calls
are supported by the work of climate historians, decolonial critics and authors who assert that
we cannot hope to advance climate action if we do not address the systems of capitalism and
colonialism that have created the current crisis and still shape responses to it (Bhambra and
Newell, 2022; Ghosh, 2022; Sultana, 2022; Yusoff, 2018). The future-focus of much scientific,
political and popular discourse around climate change can create a disconnect with the past,
occluding the fact that climate change and its associated crises 'are deeply rooted in history'
(Ghosh 2022, 158). In this context, there is a danger that the language of tipping points can be
used to reinforce a discourse that abstracts climate change from past inequities and local
contexts. The notion of tipping points that are rooted in a biophysical framing, which assumes
some 'threshold' and 'set of shocks' that tips a system over, ignores the grinding every-day
realities of life that many of the poor and most vulnerable endure as an interconnected set of
social, economic and environmental crises (Nixon, 2013). These vulnerabilities will only be
compounded by the increased risks associated with unmitigated climate change, biophysical
pressures, and tipping points (O'Brien and Leichenko, 2000).
Moreover, a focus on preventing negative tipping points can distract attention from the deep
structural imbalances of capital and the asymmetric power that both drive tipping and the
precarity and increased vulnerability to the impacts of tipping events in poorer regions (Roberts
and Parks, 2006). The urgency that accompanies the notion of tipping points can overshadow
the slow process of rebuilding trust and relationships that have been broken through past
harms, referred to by Kyle Whyte as "relational tipping points" (Whyte, 2020). For many
Indigenous peoples and local communities who have faced the existential crisis of colonialism
and who are now at the forefront of the climate crisis (Gilio-Whitaker, 2019), relational tipping
points may have already been breached (Whyte, 2020, 2021). The process of rebuilding
consent, trust, accountability, and reciprocity—qualities of relationships necessary to avoid
further injustices—require time and commitment (Whyte, 2020). Attempts to avoid tipping points
through geoengineering, for example, could merely pass on costs and irreversible effects onto
future generations (Biermann et al., 2022), while contemporary drives to reach technological
tipping points, such as the push towards electric vehicles, can produce new vulnerabilities for
communities situated in areas that are rich in rare Earth minerals (Calvão et al., 2021). Hence
without due care, attempts to address tipping points, while important, can also perpetuate
spatial and temporal inequities and injustices (Sovacool et al., 2022).
In this paper, we discuss considerations of ethics, equity and justice in relation to the complex
interconnection of biophysical and social, 'positive' and 'negative' tipping points. The
destabilising of critical Earth systems is already contributing to adverse effects on human well-
being and the global ecosystems on which it depends, and will continue to worsen (Rockström
et al., 2023). Crossing biophysical and social tipping points will exacerbate current injustices
and inequities (Rammelt et al., 2023), as well as increasing potential harms on future
generations and limiting their response capacity by triggering potentially irreversible processes.
It is thus necessary to approach PTPs with due precaution and humility in our understanding of
how complex social-ecological processes unfold- as such we refer to the need for an ethics of
tipping points interventions that centres considerations of equity and justice as central tenets.
**1.2. Discourse matters**
Within the framework of tipping points, it is crucial to remember that all human and non-human
actors (sometimes referred to as more-than-human actors) are, in Donna Haraway's words,
'situated.. in complicated histories' (Haraway, 2016), which inform complex and plural visions for
the future. The IPCC AR6 report urges immediate action and deep emissions reductions in this
decade whilst also calling for climate resilient development that prioritises risk reduction, equity
and justice (IPCC, 2023). In seeking to build a majority of people in favour of stronger, faster
action, it is vital that values-inclusive forms of discourse are identified to 'create a sense of
collective responsibility and action' (Wiedmann et al., 2020).
The challenges and tradeoffs inherent in achieving a safe and just operating space for life on
Earth need to be understood (Gupta et al., 2023a). Dominant discourses that centre efficiency
and technocratic solutions must shift towards ones that instead aim to reconcile the need to
meet the internationally agreed temperature targets with the need to address over-consumption
and inequalities within and between nations (Constantino and Weber, 2021; Hickel and Kallis,
2019; Lamb et al., 2020; Steinberger et al., 2020; Wiedmann et al., 2020). A growing
understanding of tipping points in the Anthropocene challenges 'the peaceful and reassuring
project of sustainable development' (Bonneuil and Fressoz, 2016: 29). We have entered what
Bruno Latour calls 'the new climatic regime' (Latour, 2018) in which the geophysical framework
that we have always taken for granted, the ground on which our history, politics and economics
have played out, has become destabilised. An ethical community of nations that respects the
Earth's biophysical limits and minimum social foundations for human flourishing must recognise
that the only viable solutions are ones that prioritise strong sustainability and sufficient access to
resources for all (Haberl, 2015; Trebeck and Williams, 2019). For example, Raworth's (2017)
'doughnut economics' has as its goal the establishment of a safe and just operating space for
humanity that includes staying above social 'floors' such that everyone has access to necessary
goods and services while also staying below the planetary boundaries,  beyond which the
economy begins to outstrip the planet's natural resources (Gupta et al., 2023b; Raworth, 2017).
This implies differential responsibilities on different groups of people as we seek to navigate
towards more just, equitable and sustainable futures.
**1.3. What do we mean by equity and justice?**
Gupta et al. (2023a,b) propose an integrated "Earth system justice" framework to approach
questions of climate justice and understand how to reduce risks associated with crossing tipping
points while ensuring well-being for all and an equitable distribution of benefits, risks and related
responsibilities. Earth system justice is conceptualised through multiple approaches and
understandings of justice including, but not limited to, intragenerational, intergenerational justice
and interspecies justice. Intragenerational justice refers to the relationships between humans
right now and includes justice between states and social groups. Intergenerational justice
examines relationships across generations, such as the legacy of greenhouse gas emissions or
ecosystem destruction by current and past generations on youth and future generations, and
assumes that natural resources and environmental quality should be shared across generations
(Tremmel, 2009). In this context, interspecies justice requires considering the rights of nature
and other species. It draws on a rights of nature discourse (Harden-Davies et al., 2020) that
also counters the idea of human exceptionalism as a lens for thinking through development
impacts (Srinivasan and Kasturirangan, 2016) and potential remedies like ecocide (Setiyono
and Natalis, 2021). Drawing on these frameworks can help us to assess the uneven impacts of
nearing Earth system tipping points, but also the differential responsibility for efforts to avoid
tipping points and the distributional and procedural aspects of positive tipping dynamics.
Within the domains mentioned above, one can discriminate between different dimensions of
justice, i.e., distributive (or equity across different populations), procedural  (how decision or
research processes are designed, who is involved), and reparative (e.g. recognition of wrongs,
restoration where possible, and compensation for negative impacts and past injustices) (Byskov
and Hyams, 2022). Such justice approaches also include recognition and epistemic justice,
which consider the value of multiple knowledge systems, especially local, Indigenous, and
unrecognised, misrecognized or marginalised groups (de Sousa Santos, 2008). Finally,
'intersectional' justice that includes multiple and overlapping social identities and categories
underpinning inequality, underrepresentation, marginalisation, and the capacity to respond (i.e.
gender, race, age, class, health) must be considered in the context of Earth system justice
(Gupta et al., 2023c). These different forms of justice are not mutually exclusive: procedural
justice may be used to arrive at restoration or compensatory payments, which can be assessed
through the lens of distributive justice. Changes related to tipping points can be analysed with
reference to these myriad justice considerations to design forward looking actions that avoid
negative impacts.
**2.  Blind Spots of intervention**

Policymakers often overlook the normative dimensions of climate policy and the possibility of
unintended social consequences (Klinsky et al., 2017; Okereke and Dooley, 2010). However, all
actors in the process – from scientists to world leaders – must take efforts to avoid creating a
situation in which  today's solutions become tomorrow's harms. This is especially true when
considering interventions designed to trigger exponential rates of positive social change, or
quick 'fixes' such as geo-engineering (Sovacool, 2021), which could have substantial negative
impacts that could be difficult to mitigate if they are not considered before a social tipping point
is reached. It is thus imperative that all actors take responsibility to acknowledge potential risks
and centre questions of justice when considering PTPs as solutions to the ongoing climate and
other social-ecological crises.
**2.1. Risks and unintended consequences of interventions to mitigate climate change**
Interventions aimed at mitigating climate change can have unintended consequences including
poorly aligned interventions that can exacerbate existing vulnerabilities and risks. A good
example of risks associated with the quest for PTPs is the transformation to a renewable energy
economy. The growth in demand for renewable energy worldwide, including for batteries and
solar panels, is increasing the demand for lithium, cobalt and other rare earth minerals (Dutta et
al., 2016). While this creates economic benefits for mining communities, it can also produce
negative ecological, economic and social impacts in the near, medium and long-term
(Hernandez and Newell, 2022; Manzetti and Mariasiu, 2015). A recent study finds that if today's
demand for electric vehicles is projected to 2050, the lithium requirements for the US market
alone would triple the amount of lithium currently produced for the global market (Rionfrancos et
al., 2023). However, lithium demand could be reduced by 92% in 2050 relative to the most
lithium-intensive scenarios by decreasing car dependency (e.g. through increasing public transit
or biking), limiting the size of EV batteries, and creating a robust recycling system (Rionfrancos
et al., 2023). Within this context, the industrial mining sector has been accused of supporting
state violence and corruption, polluting ecosystems (Banza Lubaba Nkulu et al., 2018), and
exacerbating poverty, while the informal mining sector is known for ignoring occupational safety
and health standards and human rights concerns (Sovacool, 2019).
Other prominent examples of unintended consequences have been documented for: a) large-
scale renewable and bioenergy projects, resulting in significant local opposition (Cavicchi, 2018;
(Torres Contreras, 2022); b) the displacement of Indigenous peoples, local communities (Zurba
and Bullock, 2020) and coastal fishers (Beckensteiner et al., 2023); c) deforestation (Kraxner et
al., 2013); d) biodiversity losses (Pedroli et al., 2013); e) competition for land and water
resources (Haberl, 2015; Tarhule, 2017); f) food insecurity (Hasegawa et al., 2018); and g) for
decarbonisation of the built environment, particularly the housing stock, resulting in health
impacts from poor indoor air quality, and fuel poverty (Davies and Oreszczyn, 2012).
An example of climate policy leading to unintended outcomes with social justice implications is
'carbon leakage' (Carbon leakage, 2023; Grubb et al., 2022). Although often difficult to measure
and distinguish from the more general offshoring of emissions due to globalisation of trade and
deindustrialisation in richer countries, carbon leakage in response to climate policy measures is
an example of a negative spill-over effect. Unilateral climate policies such as carbon pricing and
emissions trading schemes (ETSs), designed to encourage carbon-intensive sectors to invest in
carbon-neutral production domestically, may lead firms to relocate to regions with equal access
to the same markets, but with fewer or less stringent regulations (Prellezo et al., 2023).
Relatedly, significant policy research has focused on the concept of a 'just transition' (Newell
and Mulvaney, 2013; Wang and Lo, 2021), spurred by the negative labour market impacts of
decarbonization measures in coal-intensive regions of the Global North (Abraham, 2017).
Unless sufficient government investment, regional regeneration, support and skills retraining are
provided to those workers and communities facing the greatest risks from a transition away from
fossil fuels, severe economic, social and cultural hardships are likely to follow. Furthermore, this
could reduce trust in government and strengthen counter-narratives aimed at delaying climate
action (Lamb et al., 2020; Patterson et al., 2018). Participatory and deliberative governance
approaches that include potential losers and other stakeholder groups in designing and
implementing policy for sustainability transitions can help to lower the barriers to a transition by
building political will and legitimacy, and negotiating effective compromises for more just
outcomes (Fesenfeld et al., 2022). More generally, climate policy needs to be designed to
subsidise lower-income households for the higher costs that may accompany measures such as
carbon pricing, emissions trading, new standards for energy-efficient buildings, smart energy
systems, and the electrification of transport systems. Failure to do so could increase poverty,
inequality, hunger and other health impacts, popular protest and political instability (Davies and
Oreszczyn, 2012; Newell et al., 2021).
In the Global South, the transition to net-zero carbon emissions must happen alongside
reductions in poverty and multidimensional vulnerabilities, and while ensuring decent living
standards for all. These countries are confronted with a toxic mix of shrinking carbon budgets,
growing inequalities, heightened climate-related risks, and limited capabilities for mitigation and
adaptation due, in part, to increasing debt burdens (Steele and Patel, 2020). But the debate on
historic responsibilities, development rights, and net-zero efforts is gaining renewed attention
(Mishra, 2021). From the perspective of the Global South, achieving just transitions requires
addressing the double inequality of the climate crisis where developing countries bear a
disproportionate share of the risks, while industrialised nations are primarily responsible for
historical emissions (Gardiner, 2004). Therefore, developing countries are demanding fair
procedures for distributing the costs and benefits of mitigation and adaptation, such as the
Warsaw International Mechanism for Loss and Damage. However, concrete financing
commitments from rich countries remained absent at COP28 in Dubai in 2023 (Jessop et al.,
296  2023).

Unpopular climate policies can sometimes trigger a widespread 'backlash' (Patterson, 2023).
Examples of climate policy backlash include the response to the Australian carbon pricing
scheme (Crowley, 2017) and the French fuel tax increase that gave rise to the Gilets Jaunes or
Yellow Vests protest movement in 2018-2019 (Kinniburgh, 2019). Other well-researched forms
of unintended impacts of policy measures include rebound effects (Chakravarty et al., 2013).

Unintended consequences can also emerge from a failure to build broad coalitions based on
value-inclusive narratives and norms (Constantino and Weber, 2021; Evans, 2017; Klein, 2015;
Meadowcroft, 2011; Rowson and Corner, 2014; Sloterdijk, 2012). Procedural justice is also key
as small producers and/or vulnerable actors are often excluded from the political processes and
negotiations that determine climate policy (Villasante et al., 2022). In centering justice and
combining multiple, intersecting social movements under the climate justice umbrella, many
campaigners and scholars believe that the strength of their combined movements can be
amplified (Mikulewicz et al., 2023). However, there are also concerns that strong social justice
framings can increase political polarisation rather than build broader coalitions (Patterson et al.,
2018; Smith, 2022). Research has also shown that some actors recognise the need for greater
urgency in climate policy, but are reluctant to champion it to avoid being labelled as 'extremists'
(Willis, 2020). As a result, climate policymakers and other actors may prefer to focus on the
more technocratic, less politically risky aspects of transition governance (Patterson et al., 2018).
If decarbonisation is left mainly to market-based mechanisms that prioritise only profitability, the
speed and up-scaling of technological change may threaten the human rights and well-being of
some people while allowing other, more powerful, incumbent actors and structures to prevail
(Newell et al., 2022). Unique opportunities to redesign entire systems and sectors along more
efficient, ethical, sustainable, and equitable lines may be lost where speed and capital
accumulation is allowed to trump inclusivity and depth of process (Leach and Scoones, 2006).
For example, U.S. solar photovoltaic deployment is forecast to grow non-linearly in the near-
term, generating around 12% of all US power by 2027 (SEIA/Wood MacKenzie, 2023). While
this is a positive development in terms of the speed of overall decarbonisation, the perpetuation
of an energy system dominated by profit-maximising utility companies would be viewed as a
missed opportunity for advocates of energy democracy and place-based, cooperative and
community-owned energy (Hoffman and High-Pippert, 2005; Stone et al., 2022). Likewise, 'plug
and play' approaches that seek to electrify cars, but not boost the accessibility of public
transport can serve to reinforce private automobility (Rionfrancos et al., 2023).
Additionally, there is a risk that a growing concern regarding Earth system tipping dynamics
could propel research into speculative interventions such as widespread carbon dioxide
removal,geoengineering or solar radiation modification—a set of hypothetical solutions aimed at
reducing incoming sunlight and thus lowering global mean temperatures (National Academies of
Sciences, Engineering, and Medicine, 2021). The most common solar geoengineering proposal
involves injecting aerosols into the stratosphere to limit the influx of solar energy, but there are
also more regional or local proposals involving different technologies. Proponents often argue
for these hypothetical solutions on the grounds that we have made little progress in reducing
carbon emissions and that solar geoengineering could be used to buy time or as a failsafe
(Keith, 2013; Keith et al., 2017). However, solar geoengineering and other more speculative
solutions often come with substantial uncertainty and risks, which are likely to vary across
regions, and insufficient governance mechanisms to equitably and effectively manage such
risks (Kravitz and MacMartin, 2020; McLaren, 2018; Schneider et al., 2020; Stephens et al.,
2021). This has led groups of scholars to call for an "international non-use agreement" and for
limits on related research as well (Biermann et al., 2022).

## 2.2. Winners and Losers: Sacrifice Zones

To include equity and justice in the discourse of tipping points, it is necessary to consider how
resource extraction can drive tipping points through resource dispossession whilst also
exacerbating the drivers leading to a transgression of planetary boundaries (Pereira et al.,
2024). Resource extraction, be it for fossil fuels or green energy sources, creates sacrifice
zones– places permanently impaired by environmental degradation and divestment- mainly in
the Global South, but also in marginalised areas of the Global North, for example, the green
energy developments in Sapmi territories in Scandinavia (Kårtveit, 2021), or lithium mining in
Portugal (Canelas and Carvalho, 2023). These actions exacerbate the transgression of
planetary boundaries (Sultana, 2023b), cutting across North and South, and are reflective of the
uneven control of production, technology and the finance which drives extractivism between
global ('polluter') elites and more marginalised social groups (Kenner, 2019).
Even well-intentioned interventions have the potential to put pressure on lands held by
Indigenous and marginalised communities and reshape their ecologies into "green sacrifice
zones" by reproducing a form of climate colonialism in the name of the energy transition (Lang,
2024; Zografos and Robbins, 2020). Climate colonialism involves ''the deepening or expanding
of domination of less powerful countries and peoples through initiatives that intensify foreign
exploitation of poorer nations' resources or undermine the sovereignty of native and Indigenous
communities in the course of responding to the climate crisis'' (Zografos & Robbins, 2020: 543).
Green sacrifice zones then are "spaces or ecologies, places and populations that will be
severely affected by the sourcing, transportation, installation, and operation of solutions for
powering low-carbon transitions, as well as end-of-life treatment of related material waste"
(Zografos & Robbins, 2020: 543). Current examples include 'green grabs' for critical minerals,
biofuels and water or the acquisition of land for forestry carbon offset projects (Fairhead et al.,
2012; Scoones et al., 2015).
The violence that capitalism inflicts on places designated as sacrifice zones can be immediate,
but it can also be slow and imperceptible. Rob Nixon describes the "slow violence" that befalls
marginalised communities over a long period of time and which is almost imperceptible in the
marking out of zones for development (Nixon, 2013). This extractive view from corporations and
governments meets the resistance of "submerged perspectives", that is, the ways in which the
local humans and nonhumans that inhabit those territories perceive life as entangled, where the
destruction of one part affects the rest of the entities and breaks the spiritual heritage in a region
(Gómez-Barris, 2017). Slow violence has delayed effects and requires justice to take new forms
to secure effective legal measures for prevention, restitution, and redress (Nixon, 2013). To
include justice and equity in climate mitigation actions, Latin American countries, for example,
have developed the first regional agreement *Acuerdo de Escazú* in 2018 (CEPAL, 2018). This
agreement proposes three concrete objectives to include climate justice in environmental
policies and transition actions: (1) access to environmental information, (ii) public participation in
environmental decision-making processes, and (iii) access to justice in environmental matters.
Such attempts  to involve communities in discussions of climate justice are crucial for an
approach to PTPs that aims to centre equity and justice frameworks. For the concept of
sustainability and just sustainable futures to address local realities, environmental justice
scholar Julie Sze argues that a "situated sustainability" is necessary (Sze, 2018). Situated
sustainability should "set the parameters for why and how vulnerability (environmental or other)
is disproportionately distributed, one of the key questions in environmental justice research"
(Sze, 2018: 13). In other words, if the questions we ask aim at transformative change or positive
tipping points, they cannot neglect how racial capitalism contributes to inequalities and
environmental degradation (Newell, 2005; Sze, 2018).
**2.3. Reinforcing current power dynamics and structures**
While averting negative biophysical tipping points in the Earth system is a global challenge that
will require a coordinated global effort, the research and policymaking surrounding positive
tipping must also grapple with historical and contemporary inequalities in the production of
environmental harms, and the differentiated and uneven capacity and responsibility to respond
or to withstand such impacts. These concerns are echoed in the principle enshrined in the
UNFCCC of 'common but differentiated responsibilities and respective capabilities' and
highlights the greater responsibility to act to reduce emissions and the likelihood of crossing
critical thresholds by richer countries and polluter elites, whether through their own direct efforts
or through the support of efforts in countries with fewer economic resources (O'Brien and
Leichenko, 2000). Refocusing mitigation attention on high-emitting groups, countries and
sectors highlights the need for interventions and policy measures that attempt to shift the
current consumption patterns of the wealthy and the actions of large private corporations
(Kenner, 2019; Newell, 2021; Rammelt et al., 2023; Wiedmann et al., 2020) and the
infrastructures of high-impact sectors such as food (reducing industrialised meat and dairy
consumption) and energy production (switching to non-fossil fuel based energy), transport
(reducing car use and air travel) and housing that, combined, comprise about 75% of total
carbon footprints (Newell et al., 2021). Furthermore, this view also highlights the need for
substantial financial transfers from the Global North to the Global South to help build climate
resilience, to compensate for irreparable losses due to climate change, and to offset the costs of
mitigation efforts (Jackson et al., 2023). Without such measures, efforts to address Earth
System tipping points risk reinforcing unequal power dynamics and current inequities.
**3.  Illustrative case studies**
**3.1 Risks and justice implications in Marine Protected Areas**
The ocean economy is expected to grow faster than the global economy in the coming decades,
reaching $3 trillion by 2030 (OECD, 2016), with well-established (e.g. fisheries, aquaculture)
and novel ocean sectors (e.g. seabed mining, ocean wave energy) multiplying their activity and
footprint in recent years (Jouffray et al., 2020). Yet, opportunities, access and benefits from
ocean interventions remain highly unequal. For instance, seafood production is highly
concentrated in a few Global North large corporations (Österblom et al., 2015), while in most
places of the Global South, the local nutritional needs are jeopardised by the activity of distant
fishing fleets, seafood trade, and the use of catches for fish oil/fish meal for animal feed (Hicks
et al., 2019). The unprecedented race for food, spaces and materials, but also the effects of
other drivers such as climate change and pollution, are exacerbating social inequities and
threatening marine ecosystems functioning and productivity. The race to occupy the oceans and
exploit more resources and at greater depths, combined with the impacts of climate change, are
leading to an increasing risk of reaching dangerous ocean tipping points (Jouffray et al., 2020;
McKay et al., 2022). Thus, there is a pressing call for transformative actions that halt and
reverse marine biodiversity loss rates (IPBES, 2019), particularly in some Global South
biodiversity hotspots.
The recent Kunming-Montreal Global Biodiversity Framework target 3 seeks to protect 30% of
the ocean by 2030 to halt biodiversity loss (30x30 target) (CBD, 2022). Through the global
Convention on Biological Diversity negotiations, conserving 30% of the ocean (and land) is seen
as an important threshold for addressing biodiversity loss and maintaining ecosystem function,
as previous levels of protection were insufficient (Baillie and Zhang, 2018; Dinerstein et al.,
2019). With Target 3 set 'to ensure and enable that by 2030 at least 30% of terrestrial and
inland water areas, and of marine and coastal areas, are effectively conserved and managed
(CBD, 2022),' it could function as a potential driver of a PTP if appropriately implemented.
However, the 30x30 target risks perpetuating historical injustices, colonial legacies and power
imbalances by imposing Western conservation models on communities in the Global South
(Obura et al., 2023). In effect, it is essential to explore the intricate social aspects of the initiative
(Sandbrook et al., 2023), offering a more nuanced and equitable discourse on PTPs in ocean
governance and conservation and the role of Marine Protected Areas (MPAs) in achieving them.
Although the positive ecological impacts of MPAs are relatively well understood (i.e. large, old,
well-enforced and 'no-take' MPAs would provide greater ecological benefits within the area
effectively protected (Sala and Giakoumi, 2018), less attention is paid to the negative socio-
economic impacts that MPA establishment can have on dependent and marginalised
communities (Bennett and Dearden, 2014; Rasheed, 2020). Past research has shown that the
MPAs can exacerbate equity issues currently present in the Global South, by further
marginalising already vulnerable coastal communities (Hill et al., 2016; Sowman and Sunde,
2018). MPAs establishment and management may exclude local and Indigenous participation,
which in turn can also lead to reduced conservation and management gains (Hill et al., 2016). A
heightened focus on increasing MPAs may entail undesirable consequences for social well-
being of vulnerable communities in a variety of ways, including forced removals and
displacement of Indigenous peoples from traditional lands and waters, loss or restricted access
rights, as well as negative impacts on food security, health, livelihoods, identity and culture
(Bennett and Dearden, 2014; Hill et al., 2016; Oracion et al., 2005; Sowman and Sunde, 2018).
Additionally, current extent and distribution of MPAs, for example in the Philippines, do not
adequately represent biodiversity, with only 2.8% of coral reef protected within no-take MPAs
(Weeks et al., 2010) or, in the context of the 11.4% of EU waters that are covered by MPAs
where 86% showed light, minimal, or no protection from the most harmful human activities, such
as dredging, mining, or the most damaging fishing gears (Aminian-Biquet et al., 2024).

A strong global focus on increasing MPAs as a 'tipping point' towards conserving marine
biodiversity, may fail to carefully and comprehensively address historical impacts and ongoing
equity issues experienced by coastal communities. In addition, measuring conservation success
based solely on a coverage metric can incentivize the establishment of large centrally-governed
MPAs (often situated in former colonies) (O'Leary et al., 2018), at the expense of relatively
small, but locally managed MPAs (Smallhorn-West et al., 2020). A looming time horizon for
30x30 may also discourage participatory and collaborative processes that may take longer to
achieve, but are more efficient in the long term (O'Leary et al., 2018). Concerning global
planning of MPAs expansion, maps are not apolitical. Global conservation planning exercises
informed by biophysical variables and cumulative human impacts placed a significant fraction of
priority areas within the Global South (e.g. Coral Triangle, Southwest Indian Ocean, Caribbean
Sea) (Jenkins and Van Houtan, 2016; Selig et al., 2014; Zhao et al., 2020), occupying the entire
Exclusive Economic Zones (EEZs) of some Global South countries (e.g. Indonesia) and thereby
perpetuating a form of green sacrifice zone. While providing important foundations, this
literature hardly discusses the ethical and governance considerations of such "conservation
planning exercises" and local socio-economics needs are either conceptualised as an extra map
layer that competes with wildlife or something to consider in future analyses.
The 30x30 initiative and the revitalization and empowerment of local communities toward PTPs
may be reconciled by balancing both biodiversity and well-being outcomes of local communities
when enhancing existing MPAs and designing new ones and seriously considering the wide
range of "other effective area-based conservation measures", including those where small-scale
actors, especially IPLCs, are empowered and included from the very beginning of decision-
making processes to enhance procedural justice (Atlas et al., 2021). Importantly, the expansion
of MPAs, across both large and small areas, should not be seen as a single strategy to balance
marine biodiversity and socio-economic needs; it must be part of a broader and more diverse
management and governance portfolio to govern our oceans in a sustainable and equitable
manner (O'Leary et al., 2018).
**3.2 Positive financial tipping points: actors and mechanisms**
In today's world, the prevailing financial ideology wields an overwhelming influence on the
course of human lives and the health of the Earth system, posing a significant threat to the
fabric of society and the environment. At the core of this paradigm lies a series of unchallenged
"absolute truths" that prioritise wealth accumulation, power, and unchecked economic growth, at
the expense of communal well-being and ecological sustainability (Fullerton, 2018). Achieving a
sustainable future leaves no choice but to avoid a transgression of planetary boundaries and
tipping points in key Earth system processes (Lenton et al., 2019; Richardson et al., 2023).
Financial actors are key players in the global economy and affect sustainability biodiversity
around the world. Several recent policy and private initiatives have been launched with the
ambition to redirect financial flows towards activities that protect natural capital, influence
ecosystems and generate equitable outcomes to people in a positive way (Galaz et al., 2015).

Large financial actors have been shown to possess significant corporate control globally
(Fichtner et al., 2017). Through their influence over economic activities that modify ecosystems
associated with tipping elements, financial actors can also affect climate stability and
biodiversity. A financial sector tipping point that reconfigures flows of finance towards climate
mitigation, adaptation, loss and damage compensation, biodiversity conservation, addressing
vulnerability etc. requires reimagining and reconfiguring governance of public and private
finance (Rammelt et al., 2023). This includes changing the mandates of multilateral
development banks, reforming central banks and regulating private company law and disclosure
policies while also addressing issues such as debt and taxation as part of a more transformative
approach to climate finance (Newell, 2024).
Higher costs of accessing finance in the Global South, for example, may mean that many
countries are unable to invest sufficiently in providing access to basic services like electricity
(Ameli et al., 2021), which underpin provision of healthcare and clean water, food security, and
access to information and economic opportunity. The most vulnerable in these countries stand
to gain significantly from the low-carbon transition, with cost reductions in renewable energy
generation making solar PV the most viable way to provide electricity to the majority of those
currently without access (nearly 600 million people in Sub-Saharan Africa alone) (IEA, 2022a).
Low investment due to the difficulty of accessing finance creates a higher risk-perception of
investment in these countries further increasing the cost of capital and leading to an 'investment
trap' that can be further exacerbated by climate impacts (Ameli et al., 2021). Interventions that
lower the cost of accessing capital, like credit guarantees and supporting growth of domestic
capital markets, can help to break out of this cycle and open up flows of finance to address
critical vulnerabilities and support adaptation.
There is an increasing call to change the core cause of failure of the financial system (Deutz et
al., 2020; Pinney et al., 2019; UNEP, 2023). At its core, the flawed design of finance rests on
the assumption that we can separate finance from the Earth system, and reduce the complexity
of our interconnected global economy into simplistic financial optimization calculations without
any consideration of equity and justice. Finance cannot be understood in a vacuum. Holistically
understood, finance is embedded in the real economy, which in turn must be understood as
embedded in and inseparable from the Earth system. Recently, there have been proposals to
envision a more sustainable and just financial system (Deutz et al. 2020; UNEP, 2022). For
example, regenerative 'capitalism' provides a new paradigm for finance in which true wealth is
not merely money in the bank. Rather, it must be defined and managed in terms of the well-
being of the whole, achieved through the inclusion of multiple types of wealth or capital,
including social, cultural, living, and experiential (Fullerton, 2018). To operationalize some of
these changes, a framework for guiding sustainable and equitable investments, and a taxonomy
of these investments is currently not universally defined. It is necessary to provide a
classification system of activities that comply with the principles of such investments, thereby
guiding capital investment decisions and development policy towards an improved sustainability
(Sumaila et al., 2021). One example is the United Nations Principles for Responsible
Investment[1] committing to responsible investment, which has been signed by 1400 signatories
from all over the world since 2015, and with 59 trillion USD of assets under their management.
In practice, this means that publicly listed companies globally need to abide by international
principles, even if the countries they operate in might be insensitive to such standards (Galaz et
al., 2015). Another example is the United Nations Environmental Programme (UNEP)
Sustainable Blue Economy Finance Principles where UNEP works with financial institutions to
incorporate environmental, social, and governance issues into business principles and financial
market practices (UNEP, 2020) and the Principles for Responsible Banking developed with the
United Nations Environment Programme Finance Initiative (UNEP FI) – a UN-private sector
collaboration that includes membership of more than 240 finance institutions, aimed to guide
banks to integrate sustainability across all its business areas and to align bank actions with
sustainability needs (UNEP, 2019).

The recent vision for a global, multi-directional and interconnected public investment to design a
new architecture of the finance system based on the application of a global and progressive tax
system on wealth and on more democratic ways of deciding how best to spend public
investments is one proposal for reform of the global financial structure (Global Public Investment
Network, 2023). In addition, Zucman (2016) suggests that there are several ways that would
help limit tax evasion and avoidance in the global economy. For example, the creation of a
global financial registry that tracks wealth regardless of where it is located, reforming the
corporate tax system so that the global profits of multinational companies are distributed where
the resources are extracted, and more strictly regulating banks that help evade taxes with lax
regulations. Although the secrecy practices afforded by tax havens hinder a precise
quantification, Fortune 500 companies are estimated to have US$2.3 trillion in offshore
accounts and capital positions. Tax havens cost governments between US$ 500-600 billion/year
in lost taxation, including an estimated loss to non-OECD economies of US$200 billion.
Individual wealth sheltered in tax havens is an estimated US$ 8-36 trillion, costing public
accounts further (Shaxson, 2019).
For comparison, financing needed to preserve global biodiversity is estimated at US$ 722-967
billion per year until 2030 (Deutz et al., 2020). In addition, the average global statutory corporate
tax rate has gone from 40% in 1980 to 24% in 2020, with an actual tax rate much lower in many
jurisdictions (Dempsey et al., 2022). This reduction in the tax rate for large companies has
already been shown to lead to increased inequality in different countries around the world, with
a higher risk in developing countries that are highly dependent on natural resource-based
exports (Banerjee and Duflo, 2020). At the national level, positioning sustainability as a tax
principle, integrating this dimension into corporate social responsibility on financial markets and
reducing the acceptability of tax avoidance can be powerful levers for generating the funds
needed for sustainability agendas (Bird and Davis-Nozemack, 2018). Moreover, reducing tax
avoidance, tackling illicit financial transfers, and reducing the debts of developing countries can
produce in many cases more governmental income than what has been identified in the
biodiversity finance gap (Dempsey et al., 2022).

---

[1] www.unpri.org/about-pri/the-six-principles

The above distortions are not simply a market failure, they signal a broader institutional failure.
Governments almost everywhere exacerbate the problem by paying people more to exploit
nature than to protect it, and to prioritise unsustainable economic activities (Dasgupta, 2021).
Therefore, another way to unlock the funding needed to reverse nature loss by 2030 as well as
the cost of reaching net zero carbon emissions by 2050 is to remove harmful subsidies that
harm biodiversity, such as in agriculture, fisheries and fossil fuel production (Dasgupta, 2021;
Sumaila et al., 2021). According to Koplow and Steenblik (2022), the world is spending at least
$1.8 trillion a year, equivalent to 2% of global GDP on subsidies that are driving ecosystem
destruction and species extinction. In other words, public money is funding our own extinction
(Dasgupta, 2021). To address this problem, Costello et al (2016) recently showed that global
governments could repurpose some or all of the roughly US$22 billion they annually allocate as
harmful fisheries subsidies to directly support fishers' incomes without incentivizing overfishing.
This funding could support business development capacities for fishers, be given to fishers as
lump sum cash transfers, or be used to develop and institute management reforms all of which
would support low-income fishers, particularly in the countries of the Global South. Likewise,
there have been proposals to redirect a significant percentage of the USD $11 million a minute
governments currently spend on fossil fuel subsidies to a Global Transition Fund to support low
carbon energy pathways in poorer regions of the world (Newell and Simms, 2020).
**4. Implications for practice**
Above we have laid out a series of risks and potential injustices associated with the need to act
quickly to address the existential threat of climate change and related sustainability concerns,
like biodiversity loss. We argue that interventions, especially concerning narratives of positive
tipping points, cannot be divorced from current injustices and inequities in the global Earth
system and should be approached ethically. Below, we set out some specific key messages for
different actors to internalise as we all seek to shift the planet onto a more sustainable and
equitable trajectory.
**4.1. Researchers**
**4.1.1. Employ inclusive and plural approaches.**
Biophysical and social system tipping points are interconnected, and do not exist in isolation
(Sultana, 2023a). Avoiding an increase of harms requires a broad set of expertise, approaches
and acknowledgment that we need multiple and plural approaches not only within academic
disciplines, but also of diverse knowledge systems beyond academia and that these need to be
taken seriously (Tàbara et al., 2022). Interactions with other knowledge systems are only slowly
developing, and participatory approaches that involve stakeholders in science can still be very
superficial and not go beyond consultation into more embedded modes of knowledge co-
production (Chambers et al., 2021; Osinski, 2021). By being more mindful about inclusiveness,
we can increase justice in research through participatory co-design, action research and
humility on the part of researchers (Huybrechts et al., 2017).
**4.1.2. Diversify expertise across multiple places.**
Science has an agenda-setting function that could benefit from accounting for the heterogeneity
of the expertise that is needed to solve complex problems like tipping points. Diversity is a key
principle of resilience and should also be a core framing when thinking through justice, so that
diverse groups, perspectives, knowledge systems and research methods are not side-lined in
the quest for addressing global tipping points. Place- and context-specific information and
experience is often lacking as traditional research is concentrated in high-income countries. A
more inclusive global research programme to reflect on the justice and risk aspects of the Earth
system and understanding the full breadth of impacts of positive and negative tipping points
needs to be undertaken. Greater diversity in research is therefore needed - in terms of cultural,
religious, ethnic, gender or background of the researcher, but also in the disciplines that are
engaged. For example, considering social sciences in the intention, design, implementation and
evaluation of interventions are also more likely to avoid harms and associated costs, with
potential to achieve both positive social and ecological impacts on people (Latulippe and Klenk,
2020). Including diverse groups, perspectives, and knowledge systems in the quest for
addressing global tipping points will enhance resilience and success for social tipping and will
broaden the type and scope of research undertaken (Stirling, 2010). To harness relevant social
tipping opportunities we need to learn about diverse living realities and interact with actors
outside science (Bentley et al., 2014). Diversity and inclusivity of research teams–within and
beyond academia– are needed to help find solutions to tipping points that do not exacerbate
existing injustices and inequalities (Latulippe and Klenk, 2020; de Souza, 2021).
**4.2. Business and finance**
**4.2.1. Transform financial systems**
Finance and business are a part of social and ecological systems and not apart from them.
Active steering and regulation are therefore required to divest, de-finance and divert financial
resources away from the drivers of unsustainability towards sectors and regions where they are
most required and where positive tipping points can be found (Newell, 2024). Transformation of
financial systems must extend to providing mechanisms to transform sufficient financial assets
back into biodiversity and climate assets held in secure commons instruments that can ensure
equitable access to all, in particular in developing countries (IPBES, 2022). This requires a
greatly strengthened architecture of global financial governance that prioritises sustainability
and social justice (UNEP, 2015). Reaching a financial sector tipping point implies changing the
mandates of multilateral development banks, reforming central banks and regulating the need to
change company law and disclosure policies. But as part of a global just transition and social
compact, issues of debt relief and reform of taxation have to be on the table to ensure positive
tipping points in the financial system that reduces rather than entrenches poverty.
**4.2.2. Introduce investment restrictions for non-compliant companies**
Financial actors, such as international development banks, institutional and private investors,
venture capital, credit rating agencies and international commercial banks, are increasingly
interested in the financial risks of climate change and associated changes in ecosystems (Galaz
et al., 2018). It is crucial that capital investments steer the sector toward improved sustainability
and PTPs, as opposed to overexploitation of labour and resources (Hickel et al., 2021) by
integrating sustainability and equity into traditional finance mechanisms (Jouffray et al., 2019),
through ESG approaches or measures like the social cost of carbon (Prellezo et al., 2023).
Cutting off investment for companies that are seen to be complicit in transgressing planetary
boundaries, such as some oil majors and powerful cattle lobby groups in the Brazilian Amazon
(Piotrowski, 2019), has the potential to reshape the business environment towards more ethical
practices. Another area where investments could leverage positive tipping points, for instance,
would be to finance a structural shift from car dependency as this could potentially ease
pressure in the mining sector, reinforcing reduced social and environmental harms and a
densification of metropolitan areas, which would experience myriad benefits from improved air
quality to pedestrian safety (Rionfrancos et al., 2023).
**4.2.3. Develop more supportive and inclusive investments**
Redirecting public and private money to innovative tools and instruments can enable new
entrants while reducing the degradation of biodiversity. With this improved and new direction of
finance mechanisms, businesses should then be able to both meet standards and operate in
vulnerable areas that need finance to become more resilient. This includes moving money to
key areas where it is needed (adaptation, biodiversity, social common goods) rather than just for
profit (Crona et al., 2021). For example, the IIX Sustainability Bonds are debt securities that can
be listed on a social stock exchange, and they explicitly address the inclusion of women in
economic activities. There are also initiatives to supplement gaps in the national currency
systems such as Community Inclusion Currencies[2] that empower communities to create their
own financial systems based on local goods and services (Ruddick, 2023). The Netherlands, for
example, provides special green investment funds that are exempt from income taxation, thus
allowing investors in green projects (e.g. green shipping, renewable energy development), to
contract loans at reduced interest rates (usually ~2% below commercial rates). Another
example is the Raven Indigenous Impact Fund[3], a new innovative financial product committed to
Indigenous-led equity investments in mission-driven and innovative indigenous enterprises to
help build a renewed and sustainable Indigenous economy in Canada and the US. The Climate
Bonds Initiative[4] has also a number of sector criteria (e.g. for marine energy and water utilities);
while other relevant initiatives include the Blue Natural Capital Positive Impacts Framework[5] and
the technical guideline for blue bonds. Mainstreaming these examples as best practice is critical
for leveraging the financial system to enable PTPs.
**4.3. Decision and Policy-makers**
**4.3.1. Design fiscal policies that are cognizant of extant configurations**
Fiscal policy needs to be designed to subsidise lower-income households for the higher costs
that may accompany climate policies such as carbon pricing, emissions trading, new standards
for energy-efficient buildings, smart energy systems, and the electrification of transport. Failure
to do so could set off a cascade of unintended consequences and increase poverty, inequality,

---

[2] https://grassrootseconomics.org/
[3] https://ravencapitalpartners.ca/investments/impact-funds
[4] www.climatebonds.net
[5] https://bluenaturalcapital.org

hunger and other health impacts, popular protest and political instability. Hypothecation, for
example redirecting funds from fossil fuel subsidies to affordable public transport or from
windfall taxes on oil companies for home insulation schemes, can build support among poorer
groups for measures that might otherwise be opposed. Policy and governance actors attracted
to tipping interventions need not only to design targeted, sector- and actor-specific approaches,
but also to combine disciplines and sectors for a coordinated, complex systems thinking
approach and capabilities. Including potential losers in the design process can also reduce
opposition and ensure more equitable outcomes.
**4.3.2. Foster anticipatory governance to account for unanticipated consequences**
While "positive" tipping interventions are appealing for policymakers by promising to initiate
rapid, significant and potentially irreversible change towards a desired state, careful deliberation
and participatory processes should be used to reach an agreement on what the desired change
is, what the associated trade-offs are, and which populations it is likely to benefit or harm. Given
the high levels of uncertainty associated with tipping point dynamics in complex systems, and
the multiplicity of possible post-tipping states, careful consideration must be given before
initiating a deliberate "positive" tipping intervention, with a focus on anticipatory governance that
seeks to imagine the potential futures that could arise and act accordingly (Olsson and Moore,
2024; Vervoort and Gupta, 2018). Interventions for transformation should be carefully monitored
to avoid unintended negative consequences and to address distributional harms that might
ensue (Olsson and Moore, 2024; Tàbara, 2024). The risk of unintended consequences that
might ensue after a tipping process has been initiated may require new governance
mechanisms or a stronger commitment to adaptive management practices anc capacities,
including a specific focus on monitoring the change process so that corrective measures can be
introduced. Accountability structures for 'tipping gone wrong' should be included in legal
frameworks in order to hold actors accountable for the impacts of their actions.
**4.3.3. Build appropriate institutions to govern non-linear dynamics**
Existing governance institutions may be poorly fit to the challenges associated with the
governance Earth system tipping points, which are non-linear, can have cascading or systemic
effects, and span long time horizons (Milkoreit et al., 2024; Pereira and Viola, 2018). Additional
research is needed to identify adequate governance principles and institutional structures to
manage Earth System tipping points, including ensuring equity and justice are centred in efforts
to prevent tipping points and efforts to respond to their impacts (Milkoreit et al., 2024). Tipping-
point governance should include lessons learnt from multi-scale, anticipatory governance (Boyd
et al., 2015), grounded in systemic risk approaches (Centeno et al., 2015).
**4.4. Media and communications**
**4.4.1. Be aware of the politics of language and power dynamics in science**
Communicators are a key actor who interpret the world and they are capable of constructing
new social realities and inspire action (Kegan and Lahey, 2001). They must be alert to the
ideologies, values and systems of power that affect which messages are communicated and
how they are encoded. For example, how a tipping point is identified (Juhola et al., 2022), what
specific language is used to define and communicate it (Milkoreit et al., 2018), and when it may
be used inappropriately in discussing solutions (Milkoreit, 2023). This is particularly relevant in
relation to the language of 'positive' and 'negative' tipping points, which can imply a universality
of effect that is insensitive to the diverse experiences (and responsibilities) of different
communities illustrated above.
**4.4.2. Recognize contested framings of key messages in the scientific landscape.**
In an equity and justice context, media and communicators must be alert to the competing
ideologies and value systems that affect how a message is 'decoded' or interpreted by different
communities (Holmes, 2020). The meaning of a message is not necessarily determined by the
messenger or the message, but 'a complex interplay of how this meaning is framed though
ideological values and beliefs' (Hall, 1980). Thus, it is important to view communication not as a
neutral process of information transmission, but as a complex, non-linear system that is
entangled with competing knowledge and powers. Studies have shown that increased
knowledge does not automatically lead to enlightened action (Norgaard, 2011) and, indeed, that
more factual information may serve to further entrench dismissive perceptions of climate change
(Bain et al., 2012). There is, therefore, a need to go beyond the linear 'information deficit'
models of communication, moving instead towards 'non-linear' models of communication that
prioritise open, reflective dialogue between different stakeholders. For example, case studies of
communication strategies involving Indigenous people and local communities on the frontline of
climate change have found that messages rooted in empirical research and using simple
language are insufficient and that researchers should investigate different stakeholders'
understandings of what good climate change communication is and through this determine
the needs of different audiences from their unique cultural standpoints (Barau and Tanko,
2018; Gotangco and Leon, 2017). With this in mind, it is important that communication
strategies are co-produced with the communities they are seeking to engage (Moser, 2016).
**4.4.3. Embrace creative co-production practices.**
Different initiatives have been arising from the Arizona State University Imagination and Climate
Futures Initiative, the University of Exeter-led 'Climate Stories' and 'We Still Have a Chance'
projects, the Rapid Transition Alliance's curation of 'evidence-based hope' and the Seeds of
Good Anthropocenes project. These have shown that the arts and humanities offer models for
empowering communities to create their own narratives and contextualise climate change in
relation to their own systems of value, which is an important step towards the design and
implementation of just and equitable transitions (Milkoreit et al., 2016; Roberts et al., 2023;
Woodley et al., 2022). The effectiveness of literature, film, theatre and art in promoting ethical
responses to climate change is increasingly being recognised in empirical studies (Houser,
2014; James, 2015; von Mossner, 2017). As David Holmes states, 'the arts have an ability to
communicate the vulnerability and sensitivity of climate issues that other channels may lack'
(Holmes, 2020). Therefore, in the context of tipping points, engaging a wide range of
stakeholders in creative co-production would offer an open-ended, non-instrumental approach
to communication that could be key to achieving ethical solutions in this complex field.

## 5. Conclusion


Biophysical tipping points pose existential threats to current and future generations, both human
and non-human, with those currently underserved being the most vulnerable. It is therefore
imperative to act. We also know positive tipping points are possible, but that any intervention
must take care not to perpetuate past and current injustices and inequities. Considerations of
what needs to transform, who is being asked to change and where the change or its impacts will
be felt and by whom, require a level of reflexivity and systemic understanding. There are
multiple potential points of intervention and strategies that can be adopted within a complex
ecosystem of transformation to help address the power inequalities, social exclusions and
governance gaps that are currently driving us towards Earth system tipping points. All actors
have a role to play in ensuring that justice, equity and ethics are centred in these interventions,
with a particular emphasis on the inclusion of those most affected by disruptive environmental
change and the least responsible for causing it. Finally, enabling PTPs towards radical
transformations will benefit from more diverse perspectives to open up the solution space,
leveraging a shift in worldviews and paradigms rather than just reconfiguring materials and
feedback (sensu Meadows 1999). Trying to fix a system using the same tools that created it is
not the way to address our planetary crises. Taking a cautious step back to explore all options,
not just those that seem to offer a quick fix or 'low-hanging' fruit, could offer a more substantial
route into thinking through what positive tipping points could create a more equitable as well as
sustainable future.

Author contribution
LP conceptualised the paper and prepared the initial draft together with SRS, LG, PN, BS and
SV. TA, AC, SC, AG, CV, TP and CZ edited and reviewed the draft.

Competing interests
The authors declare that they have no conflict of interest.

Acknowledgements
LP is funded by the Future Ecosystems for Africa Programme in partnership with Oppenheimer
Generations Research and Conservation and by the Swedish Research Council FORMAS
Project No 2020-00670. SV gratefully acknowledges the financial support from EQUALSEA
(Transformative adaptation towards ocean equity) project, under the European Horizon 2020
Program, ERC Consolidator (Grant Agreement # 101002784) funded by the European
Research Council. TP is funded by the Bezos Earth Fund and the Oppenheimer Programme in
African Landscape Systems in partnership with Oppenheimer Generations Research and
Conservation.

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
