# Peer review of "Beyond tipping points: risks, equity and the ethics of intervention"

_EGUsphere, 2023_

## Author Response (AR1)

**Response to reviewers**

**Dear Reviewers**

**Many thanks for your deep engagement with the paper, we have gone through all your comments and reworked the entire paper to make it clearer and also to substantiate with much more evidence. We hope that the argument now very clearly flows from the analysis and that the manuscript is now much stronger and more coherent.**

Reviewer 1
General comments: The authors provide an interdisciplinary assessment of considerations related to justice, ethics and equity in the context of (social) interventions aimed at addressing existential threats to present and future generations (i.e., climate change, biophysical tipping points). The topic is a critically important one well-deserving of broad attention and the authors do a nice job of highlighting and summarizing many relevant concerns and considerations in this complex area. Although my overall assessment of the paper and its aims is generally positive, I also believe the authors have more work to do before the paper can fully achieve those aims and be particularly impactful for a broad readership. In particular, there are two areas for revision that I would encourage the authors to pursue—they are related to one another but also slightly distinct.

The first is with regards to the empirical evidence base that supports the entire manuscript and the authors' core claims/arguments (e.g., "trying to fix a system using the same tools that created it is not the best way to go about solving our planetary crises"). In its current form, the manuscript simply lacks sufficient references to empirical research to back up many of the claims made in the paper. Many claims are simply stated as true and seemingly unquestioned (sometimes with citations provided, but not always) before the authors move on to another idea or argument. How do we know, for example, that trying to fix a system with the same tools that caused existing problems is the best way to solve global-scale crises? This is a statement that requires at least some sort of evidential support. Given that the authors are taking and calling for interdisciplinary approaches, it would make sense to draw on a wide variety of literatures and types of evidence to backup the various claims made throughout the paper. As it currently stands, I think it's likely difficult for non-expert readers to confidently assess the strength of the authors' arguments (including the entire section on what can be done better) in large part because there isn't a ton of reference to existing empirical work on these and related topics. This diminishes the likelihood that the manuscript will be impactful in the ways the authors seem to hope it could be (e.g., informing/changing practice on-the-ground; shifting research approaches to be more inclusive). Also, where there is a lack of empirical research to back up claims (particularly non-normative ones), the authors should at least be transparent about this and let readers know that claims are being made in the absence of strong supporting evidence.

**We have added a substantial number of references to back up the arguments made and are then clear about our argumentation when there is a lack of research into the topic**

The second issue is that, in addition to needing much more thorough reference to (and actual explanation of) the empirical evidence base that supports the paper's claims, the paper is largely written at such a high level of abstraction that many readers (particularly non-experts) may not really understand the nuances of what the authors are claiming or arguing for. I think this lack of concrete details is present throughout the current manuscript but is particularly problematic in section 4. As noted in my line-by-line comments below, the paper (and section 4 in particular) would be *much* stronger if the authors provided illustrative examples that help readers understand what, e.g., a "non-linear model of communication" actually looks like in practice (ideally in this domain/context). Yes, this will make the paper quite a bit longer, but it would also bring the various concepts and arguments "to life" in a way that I think would be very beneficial for many readers. Perhaps this can be accomplished through adding super brief discussion of mini- or 'micro-' case studies to each of the relevant sections (again, particularly each subsection of section 4); alternatively, maybe there are 1-2 larger, more complex case studies that the authors could draw on to highlight and explain some or many of the points they make in that section of the paper, again, in the service of helping make things much more concrete for readers.

**A much more concrete explanation is now provided, with reference to examples to help concretise the arguments. However, as pointed out, space is a constraint, so this had to be done in moderation.**

Revising the manuscript in line with those two general points of feedback will, I think, make for a much more robust and effective paper overall. A few more detailed points are provided below.

--

99: "values-inclusive forms of discourse" — would be helpful to explain and provide examples of phrases like this throughout the paper for the diverse readership

**We are now clearer throughout the paper and try to avoid jargon**

170: "It is thus imperative that all actors take responsibility to include a justice framing, acknowledging potential risks, when referencing positive social tipping points as solutions to the ongoing climate and other social-ecological crises." The authors state a strong claim here but it is kind of subtle or hidden in some ways. namely, that a justice framing somehow guards against the potential for unintended negative consequences as a function of rapid change (whether it happens as a result of intentional action or is a consequence of climate or other impacts). It seems really critical that they spend much

of the rest of the manuscript defending this proposition and highlighting (ideally with lots of diverse and strong evidence) the mechanisms/pathways through which justice framing will serve the proposed role.

**We clarified our language, both in this claim and throughout the manuscript, to pull back on the claim as well as better correlate it to the magnitude of examples. As with the answer to above, much more evidence is used to substantiate the claims being made**

240: evidence regarding rebound effects in carbon-related decision-making (at the individual/household level, at least) is fairly clear at this point and suggests that they are not much of a concern in real-world settings (i.e., actual observed rebound effects are relatively minimal in this domain). I would strengthen the "the importance of rebound effects is contested" since they really don't seem to be a meaningful problem/concern in this context. This doesn't mean that the last sentence in that paragraph (243-245) is wrong, just that purported rebound effects aren't strong support for that larger claim.

**The paragraph has been edited to reflect the negligible observed impact of rebound effects, while still including them as a helpful reminder of the complexities and uncertainties of complex systems**.

450-466: section 4.2 contains a number of grammatical/sentence structure issues that should be addressed. Some of the sentences are incomplete (structurally) and need to be revised.
**The paper has been read through and fully edited**

498-501: a number of sections, including 4.4.1 are too vague in their current form to provide sufficient guidance to many readers. Even one illustrative example per recommendation (e.g., micro-case study) would greatly add to the recommendations section. The following section (4.4.2), for example, makes some important recommendations but only expert readers will know what the authors mean when they speak of, e.g., "a non-linear model of communication…an open, reflective dialogue" and this is another place where a concrete example (or at least pointing the reader to successful efforts that follow this model) would really add a lot.

**This section in particular has been edited for clarity and concrete examples are now given.**

535-538: "Trying to fix a system using the same tools that created it is not the best way to go about solving our planetary crises. Taking a cautious step back to explore all options, not just those that seem to offer a quick fix or 'low-hanging' fruit, could offer a more substantial route into thinking through tipping points that could create a more equitable as well as sustainable future." Although I am quite sympathetic to this concluding statement personally, I'm not sure the authors have really provided sufficient evidence to support it (particularly the first sentence) throughout the present manuscript. Others may reasonably disagree with my assessment here, but I would encourage the

authors to try and point to very concrete pieces of evidence from the very diverse disciplinary, interdisciplinary and transdisciplinary literatures on these issues to back up the underlying and critically important (and deeply challenging) claim that transformational change cannot come from within the problematic or "broken" system but instead requires new tools that come from outside the system. I also was a little confused by the call for "a cautious step back" in this final line, as this seems sort of at odds with the main thrust of the rest of the paper.

**With the specific evidence now presented, we hope that clearly becomes the logical conclusion of our argument. Also, the reason for advocating for a cautious step back regarding interventions should be much clearer in terms of taking time to unpack the various unintended consequences of interventions.**

**Reviewer 2**

This is an ambitious paper about a topic – social tipping points – that is very much needed.  The collaborative team of authors brings a diversity of perspectives and backgrounds that is important.

For the paper to have the impact that it strives for, however, revisions are needed to make the critically important points more clear and understandable to a general audience.

Overall, the paper in its current form is a bit muddled.  It has potential to be a very strong paper but it is not there yet.

The most basic point is that the concept of tipping points is not adequately introduced and conceptualized.  The paper begins by acknowledging both biophysical and social tipping points – but similarities and differences between these two are not clearly explained.  And the definition of what a social tipping point is – that it is "deliberate interventions into systems with the expectation of non-linear impacts and widespread change" is a narrow definition that may not resonate with all readers. Social tipping points can result without deliberate interventions – historically we see examples of widespread social change that occurred without any specific deliberate intervention that *intended to* catalyze the resultant change.  So for a paper on tipping points, please be careful in your definition and make sure that you are inclusive different kinds of social tipping points – not just those imagined by activists who want radical social change.

**We have thoroughly worked the introduction to define and conceptualise tipping points and how we apply the term in the paper. This also aligns with the other papers in the special issue.**

Also, distinctions between positive and negative tipping points are intermingled into the discussion (and the title) without a clear framework for understanding what exactly counts as a tipping point.  One fundamental problem that the paper explores (but in a

way that is a bit confusing) is that tipping points are points of no return but the outcomes on the other side of the tipping point is generally unknowable, or at least highly uncertain.  Unfortunately, the authors seem to refer to tipping points in the same way that the socio-technical transitions literature refers to transitions – as if the directionality of the social change is clear.  But with tipping points – unlike with some of the transitions literature that is clearly conceptually grounded in a change in a certain direction (i.e. toward sustainability or away from fossil fuels to renewables) - the concept integrates a deep sense of uncertainty and chaos on the other side.  Yet the authors seem to refer to tipping points at some points in the paper as if they were talking about managed transitions. While tipping points might be anticipated, they are often not "managed".

**We have now clearly stated the definitions that are working with and delineated how we differentiate positive and negative tipping points, as we agree, these were getting intermingled. We have also clearly differentiated from the transitions literature, but maintain the link to the transformations literature that is conceptually much more aligned with the non-linear, non-manageable dynamics than the transitions literature.**

Specific Comments

1. TITLE: Risks, Ethics and Justice in the governance of positive tipping points

This title is not compelling.  The title would be more accurate if it mentioned social tipping points rather than positive tipping points.  My sense is that we need more critical analysis of the concept of social tipping points – but we donnot need confusion regarding defining a tipping point, whether it be physical or social – as positive or negative.  That is simplistic and counter to one of the main messages of the paper – that any change has unintended and harmful consequences to some so there is no such thing as a tipping point that would be ALL positive for everyone everywhere.  I always suggest putting the most novel and compelling idea at the front and center of the title – so the title should begin with tipping points – and then describe risks, ethics and justice if that is what the authors want to prioritize for their potential readers.

**This is an important point and we have decided to rephrase the title to 'Beyond tipping points: risks, equity and the ethics of intervention'**

1. Conceptual clarity – definitions and clear descriptions – are needed for a few key concepts including the concept of positive tipping points and the idea of governance of tipping points.

Positive tipping points?  I am skeptical whether tipping points can be considered positive or negative.  Given that all tipping points have both negative and positive impacts, how and why do the authors feel the need to define positive tipping points?  It seems counter to one of the main points of the paper regarding justice and equity of consequences and impacts of social change.

**We have added clear definitions based on the latest Earth science and social science literatures. We have also added a clarification that this does not relate to**

**mathematical relationships in the traditional sense. It is true that even positive social tipping points (following our definition of enabling transformation to stay within safe and just Earth System boundaries) might not be regarded as positive by some and that some might be impacted negatively by the transformations. Yet, largely, safeguarding human wellbeing as a core of the motivation for positive social tipping, can be defended in our opinion.**

Governance of Tipping Points – Tipping points are a threshold of change.  In many cases, that threshold is unknowable exactly when the system will meet that point of no return.  So how and why should we try to define and discuss and characterize governance of tipping points? This uncertainty is clearly one of the key characteristics of tipping points – and how to govern before the tipping point versus what governance should look like after the tipping point might be very different.  So I would suggest that the authors allow for more nuance and a broader scope in how the define and explore governance of tipping points.

**We have nuanced the reference to governance throughout the paper, pointing out that the paper is not arguing for how to govern tipping points broadly, but focussing in on the very specific governance issue of equity and justice. We agree that uncertainty in particular is of utmost importance to the governance of TPs, but that would probably need its own paper.**

1.  References - Some sections have no reference to the existing literature in the area. For example, section 4.3.2 on moving money where it needs to go has no references. This is a burgeoning area of research, and the authors do a disservice not to mention the recent research in this space.  Below are a few suggestions although there are many more references that could and should be added to this section.

**In order to streamline the paper, we have now collapsed 4.3.2 into 4.2.2 and included the key references that stood out for us in this space. We improved and expanded this section by addressing how moving money to new financial initiatives could help to promote sustainable and equitable finance. We provide specific examples of powerful initiatives currently in place. We also include not only several relevant references from the existing literature on finance and biodiversity, but also examples of new initiatives towards sustainable finance.**

2.  Abstract – The second sentence says that social tipping points have potential to address the challenges of biophysical tipping points – but these two sentences are not well aligned. Third sentence says talks about "the imperative to act" but it is not clear what imperative the authors are speaking about. The fourth sentence talks generally about change – and social justice within change processes – but it does not explicitly connect with the idea of a tipping point. The rest of the abstract is about inclusivity and diversity of perspectives and participatory processes in radical transformation – but this does not seem specific to the idea of tipping point.  This could be more generally applied to the transition literature or other

discussions about transformation.  For the paper to hold together well, I would suggest that the authors have to situate tipping points within a larger transformation literature and conceptions.  Connecting more explicitly to the existing transitions literature – and distinguishing from this if there is something different to be said when considering tipping points as compared to more general transformation.   This should happen in paragraph 1 of the intro where the idea of social transformation is just assumed to be linked to social tipping points.  But how these two ideas are connected is not explained.

**A much stronger link to the transformations literature has been included, differentiating tipping points from transitions**

3.  Section 1.2 – excellent point about how the idea of impending biophysical tipping points instills urgency that results in reactions that reinforce legacies of spatial and temporal injustice.

**Thank you**

4.  Section 4.2 is underdeveloped. There is so much more that could be included in terms of business and finance.  Refer to recent work on financialization, monetary policy and climate justice.  Not clear whether these are recommendations for people in business and finance or recommendations about those involved in money and finance.  If it is the later – who specifically are these recommendations for?  Policy-makers who can regulate banks?  The recommendation to have more supportive and inclusive investments (Section 4.2.2) is naïve and unsophisticated.  Who is this recommendation for?  What sector are the authors talking about?  There is so much more developed work in this space and the authors don't seem to acknowledge it.

**Unfortunately, space is a constraint, but we do agree and have tried to substantiate this as much as possible. We cannot cover everything in this space as it is not a review, the same could be said for all the recommendation sub-sections: that they could be entire papers themselves. We have combined 4.3. Into this section and included more references. We have also acknowledged that there is more out there that we don't frame in this particular paper.**

5.  Conclusions – the authors claim that it is the tipping points that are so dangerous. But biophysical change could be slow and gradual without tipping points and still be super dangerous so one could argue that the tipping points are not what is dangerous.  Similarly, with social change – the mechanism for the change, i.e. when and if there is a tipping point, seems less important than the direction of the change.   Tipping points are just one way to describe a radical abrupt change but can big change happen without tipping points?   And that more slow and steady change is also very important.  I guess this is a caution for the authors not to be so focused on tipping points when many of the important points they are making are relevant to all social change – whether or not it is a tipping point.

**We fully acknowledge the reviewer's point, but as this is an SI on tipping points, we do need to focus on these dynamics. We do acknowledge that these are not the only dangers, however and that incremental change can be just as harmful.**

Technical Corrections

Paragraph 2 – starts talking about positive tipping point without defining it.  What is it?

Talks about positive or negative tipping points?  What are they?  They need to be defined.

**Thank you for pointing this out. We have added several sentences explaining the definitions we use in this article, distinguishing between positive and negative social tipping points and interventions.**

Talks about "operationalizing a tipping point" – what does that mean?

**We have clarified the text reflecting the above introduced definitions.**

Sacrifice zones – the introduction of sacrifice zones in paragraph 2 is a bit out of context. Can the authors provide an example about how sacrifice zones and tipping points connect?

**We have introduced sacrifice zones for extractivism in connection to tipping points by describing and exemplifying how natural resource extraction is pushing local tipping points, exacerbating emissions and pushing the Earth system and local peoples into harm.**

Section 2.1 the subheading mentions "positive interventions"  What does positive mean? Not clearly defined.

**Rephrased to interventions**

The structure of the paper needs some attention – some of the sections are not clear. Section 4 is called "What does this mean in practice?"  This title does not seem like a helpful section header.   How about something more descriptive, i.e. Recommendations for Diverse Constituents

**The overall structure of the manuscript and flow of logic has been edited and the sub-headings revisited**

---

## Author Response (AR3)

**Dear Editor**

I would like to thank the authors for revising their manuscript in response to the second round of reviewer comments. Having carefully read the latest version of the manuscript and the feedback provided by all reviewers across the two sets of reviews, I believe that some of the concerns raised have not yet been addressed sufficiently. Further (minor) revisions are necessary to address the following issues.

My main concern is with the financial system 'illustrative case study'. The discussion of the financial system has now been moved to this section of the manuscript – an option I had suggested – but the text covering three pages continues to be a sprawling discussion of problems of the financial system from a sustainability perspective and several proposed solutions at a rather general level. There is no clear focus on ethical or justice issues throughout – the link is weaker in some paragraphs than others, no clear logical structure to this section, and no clear link to PTPs. At a minimum, this section needs to be shortened with a clear focus on ethical and justice implications not of the current system overall, but of interventions related to a PTP agenda, i.e., the topic of the paper. Ideally, your definition of equity of justice – Gupta's Earth system justice – should be applied in both case studies, i.e., the case discussions should illustrate how existing or proposed or existing measures advance specific dimensions of justice or create new ethical dilemmas.

Response: **The case study of the financial system has been substantially overhauled and edited to focus on blind spots in the financial system and weaknesses in the current governance architecture and how alternative and neglected approaches have the potential to generate positive tipping points which simultaneously direct finance away from polluting activities and towards groups and regions most in need of finance. We now draw on the Gupta et al (2023) framework to ensure both justice dimensions and a focus on positive tipping points runs through this section.**

Second, the 'Implications for practice' section does not always identify the specific actors addressed. For example, individual researchers vs. research institutions and funders, any actors related to finance, fostering anticipatory governance (who?), and 'communicators'.

Response: **We have endeavoured to be more specific in noting the specific actors, not just sectors and have refined the writing**

Third, the whole manuscript would benefit from careful editing, considering (i) which arguments are needed for the paper and which ones merely related (esp. in the section on unintended consequences), (ii) the need for concision, and (iii) the placement of arguments in the manuscript to avoid repetition.

Response: **5 of the authors went through with this explicit aim in mind and made substantial reductions and we hope that the paper is now much more streamlined.**